# AI-driven 3D CT imaging prediction model for improving preoperative detection of visceral pleural invasion in early-stage lung cancer

Wakako Nagase[1☯], Kazuharu Harada[2☯], Yujin Kudo[1*], Jun Matsubayashi[3], Ikki Takada[1], Jinho Park[4], Kotaro Murakami[1], Tatsuo Ohira[1], Toshitaka Nagao[3], Masataka Taguri[2], Norihiko Ikeda[1]

**1** Department of Surgery, Tokyo Medical University, Tokyo, Japan, **2** Department of Health Data Science, Tokyo Medical University, Tokyo, Japan, **3** Department of Anatomic Pathology, Tokyo Medical University, Tokyo, Japan, **4** Department of Radiology, Tokyo Medical University, Tokyo, Japan

☯ These authors contributed equally to this work.
* ykudo@tokyo-med.ac.jp

## Abstract

Visceral pleural invasion (VPI) is a critical prognostic factor in early-stage non-small-cell lung cancer (NSCLC), significantly affecting patient outcomes. Conventional computed tomography (CT) often fails to diagnose VPI accurately. This retrospective case-control study evaluated the efficacy of artificial intelligence (AI)-assisted three-dimensional (3D) CT imaging for predicting VPI in 556 patients with clinical stage 0–I NSCLC who underwent complete surgical resection. Patients with tumors > 4 cm, those not adjacent to the pleural surface, or with unsuitable CT scans were excluded. Radiological features were analyzed using AI software capable of 3D imaging and characterization of pulmonary nodules (Synapse Vincent System, Fujifilm Corporation, Japan). The dataset was divided into training (n = 408) and test (n = 148) cohorts. Stability selection identified "Solid nodule" and "Pleural contact" as key predictors. Logistic regression analysis using these features developed prediction models for VPI. Receiver operating characteristic analysis showed that the area under the curve of the derived model was 0.831 and 0.782 in the training and test cohorts, respectively. The sensitivity and specificity were 0.739 and 0.657 in the test cohorts. These findings suggest that AI-enhanced 3D CT imaging significantly improved the preoperative prediction of VPI in NSCLC, supporting AI's integration into diagnostic processes.

## Introduction

Visceral pleural invasion (VPI) is a known poor prognostic factor in early-stage non-small cell lung cancer (NSCLC). Reports indicate that the 5-year survival rates for patients with VPI are significantly worse than those for patients without VPI in tumors

**Data availability statement:** The anonymized analysis dataset underlying the findings of this study is deposited in the institutional repository of Tokyo Medical University (https://tmu.repo.nii.ac.jp/records/2001010). Additional supporting information (e.g., variable dictionary or analysis code) may be made available depending on the repository's format and submission requirements.

**Funding:** Norihiko Ikeda reports receiving a research grant from the FUJIFILM Corporation for the Department of Surgery, Tokyo Medical University. The funders had no role in study design, data collection and analysis, decision to publish, or preparation of the manuscript.

**Competing interests:** Norihiko Ikeda reports receiving a research grant from the FUJIFILM Corporation for the Department of Surgery, Tokyo Medical University. Although we have utilized the AI software integrated within the Synapse Vincent System (Fujifilm Corporation, Japan) for this study, it is essential to note that this is a development model and not a commercially available product. We have no commercial conflicts of interest to disclose. This does not alter our adherence to PLOS ONE policies on sharing data and materials.

of 3 cm or less [1,2]. The incidence of lymph node metastasis is correlated with VPI [3]. Its prognostic value is reflected in the T category of the TNM classification for lung cancer [4]. In the era of increasing use of segmentectomy for small-sized NSCLC, accurate preoperative VPI identification is critical, as lymph node metastasis generally excludes eligibility for segmentectomy and intraoperative nodal assessment can be difficult, especially for nodes within or near the preserved segments. Accurate preoperative identification of VPI is essential for appropriate surgical decision-making and has a significant impact on treatment strategies [5,6]. Therefore, reliable VPI prediction methods remain a high priority in NSCLC management.

Radiological assessment using computed tomography (CT) has identified features correlated with VPI, including pleural tags or indents, tumor dimensions and composition, and pleural thickness [7–16]. However, these evaluations have been limited by the subjective nature of manual interpretation and the limitations of two-dimensional (2D) images. This can result in the underestimation of tumor characteristics, either because of the inability to reproducibly measure each characteristic depending on individual measurements or because 2D images do not assess the overall image of the tumor. Recently, we reported a study that utilized artificial intelligence (AI) analysis on 2D CT images, specifically focused on VPI [17].

This study aimed to investigate the use of AI with three-dimensional (3D) imaging to predict VPI using preoperative CT imaging. Recently, AI has significantly contributed to advancements in the medical field [18], assisting in highly precise evaluation of radiological diagnoses and providing objective radiological findings [19,20]. In addition, the reconstruction of CT images to 3D images enables the evaluation of irregularly shaped tumors. Studies have shown the efficacy of AI software integrated into the Synapse Vincent System (Fujifilm Corporation, Tokyo, Japan) [17,21–24]. This technology uses AI to automatically identify tumors and recognize patterns, providing both quantitative and qualitative assessments of radiographic characteristics. Nevertheless, no studies have investigated the accuracy of AI analysis of 3D CT images for VPI prediction. Therefore, we aimed to determine the ability of AI-assisted 3D CT imaging to predict VPI.

To achieve accurate predictive modeling, we focused on developing and validating an AI-driven VPI prediction model using 3D CT images. This study aimed to demonstrate how advanced AI techniques can facilitate a more accurate prediction of clinical outcomes in early-stage NSCLC, thereby contributing to improved patient management and treatment strategies.

## Methods

### Patient cohort

This was a retrospective, single-center, observational study. A total of 556 patients with clinical stage 0–I NSCLC who underwent complete surgical resection at Tokyo Medical University Hospital between January 2011 and December 2018 were included in the study. We focused on tumors located near the pleural surface, as identified on preoperative thin-section CT scans, to predict VPI in lung cancer. The data were accessed for research purposes on 21/02/2023, and the patients were

assigned to the training and test cohorts at a ratio of 3:1 based on the date of surgery. The training and test cohorts comprised 408 and 148 patients, respectively (Fig 1). Our institutional review board approved the study protocol and waived the need for informed consent, as the study involved retrospective analysis of clinical data collected during standard care (study approval no: SH3951). Detailed inclusion and exclusion criteria are provided in the Supplementary S1 File.

## Radiological evaluation

High-resolution chest tomography (HRCT) images of the entire lung were acquired as previously described [25]. The diameter of each tumor was meticulously measured by a team of thoracic surgeons to ensure accuracy. The measurements were approved at a multidisciplinary team conference involving radiologists, thoracic surgeons, and oncologists. The CTR (%) was defined as the maximum dimension of consolidation in the lung window setting divided by the maximum dimension of the tumor in the lung window setting. Detailed information and a description figure (S5 Fig) of the 22 AI-derived radiological features used in the analysis is provided in the Supplementary S1 File.

## Artificial intelligence analysis with Synapse Vincent

The AI software Beta Version of the Synapse Vincent system (Fujifilm Corporation, Tokyo, Japan) was used for analysis. In our previous work, we reported an AI analysis of pulmonary nodules using Synapse Vincent [17,21–23]. This segmentation algorithm is based on a 3D convolutional neural network using a modified U-net architecture. The system automates the detection and extraction of pulmonary nodules across the entire lung field, identifying and calculating the volumes of Ground-glass nodules (GGNs) and solid lesions along with their respective proportions. GGNs and ground-glass opacities (GGOs) are used synonymously, defined as radiological features characterized by a misty increase in attenuation that did not obscure the underlying lung structures in the lung window setting on CT scans [25]. The system automatically calculates 17 three-dimensional radiological parameters, including the maximum diameter of the tumor, solid component,

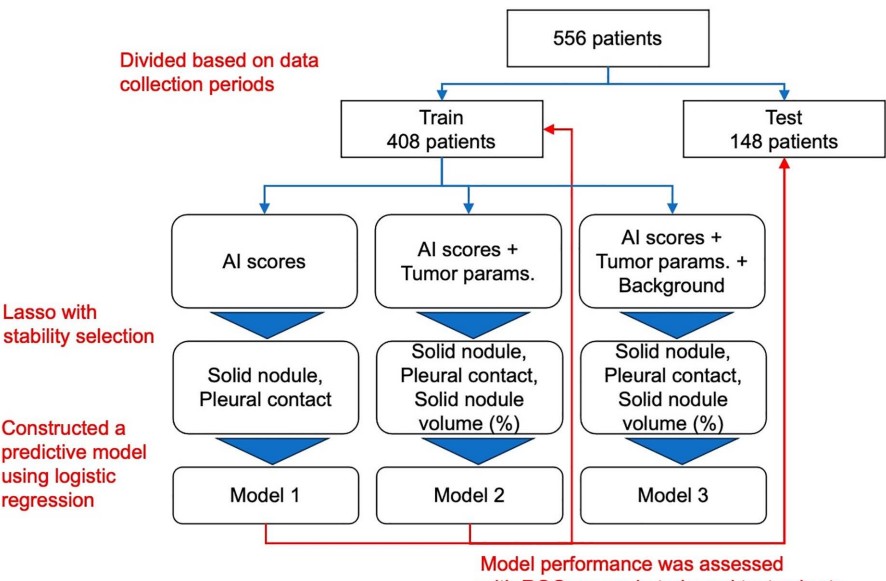

**Fig 1. Flow chart.** Out of the total 556 patients, 408 patients, were assigned to the training cohort, and the other 148 patients were assigned to the test cohort at a ratio of 3:1 based on the date of surgery (June 1, 2017). For model construction, three sets of candidate variables, including the AI confidence scores, other tumor parameters, and patient characteristics, were prepared. Models were constructed using variables selected from each candidate through the stability selection. The model's performance was evaluated using the test cohort.

and CT values. In this study, the following 3D radiological parameters were used: whole tumor volume, GGN-part volume, GGN-part ratio, solid-part volume, solid-part ratio, maximum solid length, and maximum whole-tumor length.

Additionally, the AI software beta version of the Synapse Vincent system facilitates the characterization of pulmonary nodules by assessing 22 radiological features and quantifying each with a confidence score ranging from 0 to 1 [26,27]. As described in the cited study, this algorithm uses a CNN model that is a simplified version of the VGG architecture with 12 three-dimensional convolutional layers and two max pooling layers [26]. A screenshot of the analytical process is shown (Fig 2). The 22 radiological features were as follows: clear boundary, irregular shape, round shape, smooth shape, irregular edge, serrated edge, spiculation, lobulated edge, polygon edge, bronchus translucency, cavity, pleural indentation, pleural contact, solid nodule, part-solid nodule, GGO, calcification, fat, bronchial convergence, bronchial compression, pleural recess, and pleural hypertrophy. Detailed information is provided in the Supplementary S1 File.

## Pathological evaluation

All resected specimens were formalin-fixed and stained with hematoxylin and eosin using routine procedures. Pathological staging was performed according to the TNM classification (7th edition) [28]. A detailed analysis of VPI, blood vessel invasion (BVI), and lymphatic permeation was performed [25,29]. Elastica van Gieson (EVG) and D2-40 staining are routinely used to evaluate histological structures and tumor invasion. VPI was defined as tumor invasion beyond the elastic layer of the pleura. The supplementary S1 File includes further details on pathological evaluation methods.

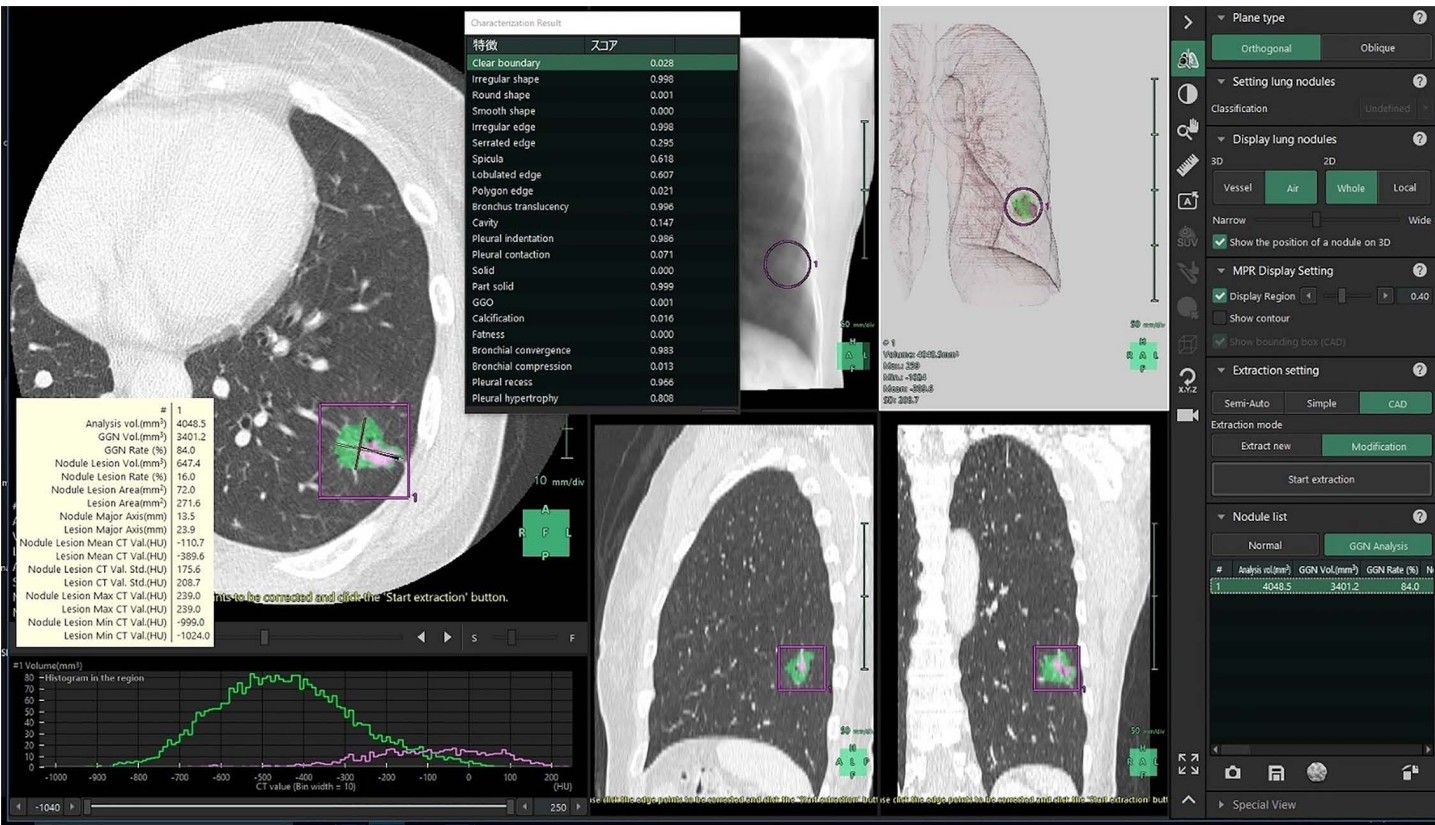

**Fig 2. A screenshot of the AI software beta version in the Synapse Vincent system.** This system automatically calculating three-dimensional radiological parameters and assessing 22 radiological features of the characteristics of the pulmonary nodule. Each feature is quantified with a confidence score ranging from 0 to 1.

## Statistical analysis

We used the Mann-Whitney U test for continuous variables and Pearson's chi-square test for categorical variables to compare the training and test cohorts and the VPI-positive and VPI-negative groups. The significance threshold was set at $P < 0.05$.

Using the stability selection algorithm with L1-regularized logistic regression (Lasso), we selected important features associated with VPI among the AI-derived features and certain patient characteristics. The variable selection algorithm was applied to three nested sets of features: AI confidence score alone, AI confidence score with 3D radiological parameters, and patient characteristics. All interaction terms were included. Using the selected features for each set, we performed logistic regression modeling to develop a VPI prediction model. Variable selection and model development were performed for the training cohort, and model evaluation was performed for the test cohort.

Variable selection methods such as stepwise regression and Lasso are highly sensitive to sample variability, and their results may not be stable. In this study, we employed the stability selection algorithm to identify factors that are likely to be associated with VPI. The stability selection algorithm repeatedly applies a variable selection method to subsampled datasets and evaluates the selection probability of each feature. This approach identifies features that are more likely to be truly associated with the outcome variable [30,31]. In our analysis, the upper bound for the expected number of false discoveries was set to one, the cutoff for the selection probability was set to 0.75, and the number of subsampling replicates was set to 50.

The AI confidence scores for the radiological features were transformed in advance. Because the original AI confidence score ranged from 0 to 1, it was slightly adjusted towards 0.5, first (let $S$ be the shrunk score), and then the logit transformation was applied as follows:

$$\log \frac{S}{1-S} \text{ , where } S = \frac{(\text{AI confidence score} - 0.5)}{1 + 10^{-8}} + 0.5$$

Logit transformation is commonly used to convert probabilities or ratios into a continuous variable that spans all real numbers, preventing values from concentrating at the ends of the original range and allowing the variable to be distributed more normally (details shown in Supplementary S1 Fig) [32].

For each model developed using the three variable sets, receiver operating characteristic (ROC) curves in the training and test cohorts are depicted. The area under the ROC curve (AUC) and 95% confidence interval (CI) of each model were calculated to compare the performances. The Statistical Package for the Social Sciences (SPSS) statistical software (version 28.0; SPSS, Inc., Chicago, IL, USA) was used for univariate analysis. For variable selection, model development, and graph drawing, we used R (version 4.2.2) and its packages stabs (version 0.6.4), ggplot2 (version 3.4.1), and pROC (version 1.18.0).

## Results

### Patient characteristics

The study included 556 patients with clinical stage 0–I lung cancer, divided into a training cohort (408 patients) and a test cohort (148 patients) (Table 1). There were no significant differences between the cohorts in age, tumor size, solid tumor size, or consolidation tumor ratio (CTR). Visceral pleural invasion was identified in 32.2% of patients, including 2.3% with pleural invasion extending beyond the parietal pleura. Lymph node metastases were present in 15.8%. Upstaging to pathologic stages II-IV occurred in 21.8% of the patients. Significant differences between the training and test cohorts were observed in age (p = 0.002) and pathological whole tumor size (p = 0.018). Detailed information is shown in Supplementary S1 Table.

**Table 1. Baseline Characteristics: Training versus Test Cohorts.**

| Variable | Overall (%) (n = 556) | Train cohort (%) (n = 408) | Test cohort (%) (n = 148) | p-value[a] (train vs. test) |
|---|---|---|---|---|
| **Sex** | | | | 0.607 |
| Women | 258 (46.4) | 192 (47.1) | 66 (44.6) | |
| Men | 298 (53.6) | 216 (52.9) | 82 (55.4) | |
| **Age, median (IQR), years** | 70 (63, 76) | 69 (62, 75) | 73 (64, 77) | 0.002 |
| **Smoking habit** | | | | 0.062 |
| Ever smoker | 340 (61.2) | 240 (58.8) | 100 (67.6) | |
| **Radiological whole tumor size, median (IQR), cm** | 2.3 (1.7, 3.0) | 2.3 (1.8, 3.0) | 2.2 (1.6, 3.0) | 0.431 |
| **Radiological solid size, median (IQR), cm** | 2.0 (1.3, 2.7) | 2.0 (1.3, 2.7) | 1.90 (1.2, 2.6) | 0.239 |
| **CTR, median (IQR)** | 1.0 (0.7, 1.0) | 1.0 (0.7, 1.0) | 1.0 (0.7, 1.0) | 0.043 |
| **Surgical procedure** | | | | 0.016 |
| Lobectomy/segmentectomy | 525 (94.4) | 391 (95.8) | 134 (90.5) | |
| Wedge resection | 31 (5.6) | 17 (4.2) | 14 (9.5) | |
| **Pathological whole tumor size, median (IQR), cm** | 2.5 (1.8, 3.4) | 2.5 (1.8, 3.2) | 2.70 (2.0, 3.7) | 0.018 |
| **Visceral pleural invasion** | | | | 0.929 |
| PL0 | 377 (67.8) | 275 (67.4) | 102 (68.9) | |
| PL1 | 129 (23.2) | 97 (23.8) | 32 (21.6) | |
| PL2 | 37 (6.7) | 27 (6.6) | 10 (6.8) | |
| PL3 | 13 (2.3) | 9 (2.2) | 4 (2.7) | |
| **Histology** | | | | 0.992 |
| Adenocarcinoma | 464 (83.5) | 340 (83.3) | 124 (83.8) | |
| Squamous cell carcinoma | 65 (11.7) | 48 (11.8) | 17 (11.5) | |
| others | 27 (4.9) | 20 (4.9) | 7 (4.7) | |
| **Pathological lymph node factor** | | | | 0.266 |
| N0 | 451 (81.1) | 328 (80.4) | 123 (83.1) | |
| N1 | 46 (8.3) | 32 (7.8) | 14 (9.5) | |
| N2 | 42 (7.6) | 36 (8.8) | 6 (4.1) | |
| NX | 17 (3.1) | 12 (2.9) | 5 (3.4) | |
| **Pathological stage (7th Ed.)** | | | | 0.250 |
| IA | 255 (45.9) | 63 (42.6) | 192 (47.1) | |
| IB | 180 (32.4) | 56 (37.8) | 124 (30.4) | |
| II-IV | 121 (21.8) | 29 (19.6) | 92 (22.5) | |

[a]Wilcoxon rank sum test; Pearson's Chi-squared test; Fisher's exact test

CTR, Consolidation Tumor Ratio; PL, Pleural invasion; Ed, Edition

## Association between VPI and clinicopathological factors

In the training cohort, men accounted for 71.4% of the patients with VPI, which was significantly higher than that in those without VPI (44.0%) (p < 0.001) (Supplementary Table S2). The median solid tumor size and pathologic tumor diameter were larger in the VPI group. The prevalence of adenocarcinoma was lower in the VPI group (70.7% vs. 89.5%, p < 0.001), and the lymph node metastasis rate was significantly higher in the VPI group (30.8% vs. 9.8%, p < 0.001).

## Relationship between VPI and AI-derived 22 radiological features

In a training set of 408 patients, the relationship between VPI and 22 radiological features analyzed using AI was demonstrated (Table 2). The "Solid nodule" was identified as the leading predictor with an area under the curve (AUC) of 0.7867,

**Table 2. Relationship between VPI and AI-Derived 22 Radiological Features in the training cohorts.**

| Variables | Overall, median (IQR) | Non-VPI, median (IQR) | VPI, median (IQR) | AUC (95% CI) |
|---|---|---|---|---|
| No. (%) with data | 408 (100) | 275 (100) | 133 (100) | |
| Clear boundary | 0.355 (0.064, 0.706) | 0.160 (0.043, 0.555) | 0.636 (0.380, 0.843) | 0.738 (0.689 - 0.787) |
| Irregular shape | 0.998 (0.988, 1.000) | 0.998 (0.988, 1.000) | 0.998 (0.990, 1.000) | 0.521 (0.462 - 0.579) |
| Round shape | 0.001 (0.000, 0.010) | 0.001 (0.000, 0.012) | 0.001 (0.000, 0.007) | 0.517 (0.458 - 0.576) |
| Smooth shape | 0.000 (0.000, 0.001) | 0.000 (0.000, 0.001) | 0.000 (0.000, 0.001) | 0.507 (0.447 - 0.566) |
| Irregular edge | 0.993 (0.958, 0.999) | 0.994 (0.958, 0.999) | 0.991 (0.961, 0.998) | 0.546 (0.488 - 0.604) |
| Serrated edge | 0.914 (0.516, 0.975) | 0.872 (0.250, 0.966) | 0.964 (0.882, 0.984) | 0.692 (0.641 - 0.744) |
| Spiculation | 0.891 (0.643, 0.964) | 0.852 (0.470, 0.956) | 0.935 (0.795, 0.973) | 0.621 (0.566 - 0.677) |
| Lobulated edge | 0.798 (0.461, 0.932) | 0.727 (0.388, 0.911) | 0.879 (0.684, 0.950) | 0.660 (0.606 - 0.714) |
| Polygon edge | 0.016 (0.007, 0.038) | 0.019 (0.009, 0.042) | 0.012 (0.005, 0.023) | 0.611 (0.552 - 0.669) |
| Bronchus translucency | 0.586 (0.085, 0.959) | 0.803 (0.212, 0.974) | 0.140 (0.035, 0.727) | 0.706 (0.652 - 0.759) |
| Cavity | 0.021 (0.006, 0.190) | 0.023 (0.007, 0.133) | 0.010 (0.004, 0.268) | 0.547 (0.483 - 0.612) |
| Pleural indentation | 0.923 (0.587, 0.981) | 0.935 (0.605, 0.985) | 0.885 (0.508, 0.963) | 0.590 (0.533 - 0.646) |
| Pleural contact | 0.705 (0.063, 0.982) | 0.318 (0.038, 0.954) | 0.969 (0.627, 0.990) | 0.704 (0.650 - 0.757) |
| Solid | 0.899 (0.001, 1.000) | 0.013 (0.000, 0.998) | 1.000 (0.994, 1.000) | 0.786 (0.741 - 0.831) |
| Part solid | 0.016 (0.000, 0.986) | 0.597 (0.001, 0.996) | 0.000 (0.000, 0.006) | 0.759 (0.710 - 0.808) |
| GGO | 0.000 (0.000, 0.001) | 0.000 (0.000, 0.004) | 0.000 (0.000, 0.000) | 0.751 (0.702 - 0.800) |
| Calcification | 0.025 (0.016, 0.047) | 0.022 (0.015, 0.041) | 0.035 (0.023, 0.071) | 0.673 (0.621 - 0.725) |
| Fatness | 0.000 (0.000, 0.001) | 0.000 (0.000, 0.001) | 0.001 (0.000, 0.001) | 0.599 (0.542 - 0.656) |
| Bronchial convergence | 0.953 (0.896, 0.979) | 0.956 (0.906, 0.980) | 0.944 (0.881, 0.973) | 0.551 (0.492 - 0.611) |
| Bronchial compression | 0.024 (0.011, 0.051) | 0.020 (0.010, 0.039) | 0.036 (0.018, 0.083) | 0.644 (0.585 - 0.703) |
| Pleural recess | 0.912 (0.755, 0.961) | 0.911 (0.743, 0.962) | 0.918 (0.801, 0.955) | 0.517 (0.459 - 0.576) |

**GGO, Ground-glass opacity**

while "Part-solid nodule" and "Ground-glass opacity (GGO)" also showed strong associations with AUCs of 0.7594 and 0.7515, respectively. Furthermore, "Clear boundary," "Bronchus translucency," and "Pleural contact" were notable for their predictive accuracy, all with AUCs above 0.7, indicating essential factors in the assessment of VPI.

## Exploration of radiological factors strongly associated with VPI, as analyzed by AI software, including clinicopathological factors

As described in the Methods section, variable selection and model development were conducted using three sets of variables: AI confidence score only, AI confidence score, 3D radiological parameters, and patient characteristics. In each analysis, the important features were first selected via stability selection. Fig 3 shows the results of the stability selection over the 22 AI confidence scores. The stability selection analysis identified "Solid nodule" as the feature with the highest selection probability, followed closely by "Pleural contact," suggesting that "Solid nodules" and "Pleural contact" are less susceptible to randomness and more likely important features for predicting VPI. In the other two variable sets, the nodule volume (%) was selected in addition to the above two features. The other results are shown in the Supplementary S2a and S2b Figs.

Next, prediction models for VPI were developed using the selected features via subsequent logistic regression analysis. Model 1 included the variables selected from the 22 radiological features, namely "Solid nodule" and "Pleural contact." Models 2 and 3 selected identical variables, including "Solid nodule," "Pleural contact," and the percentage of the solid nodule volume. Table 3 displays the logistic regression analysis results and the receiver operating characteristic areas

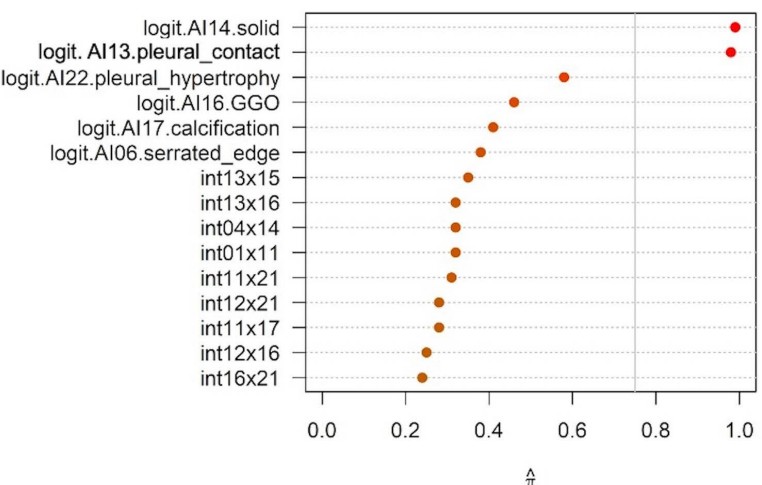

**Fig 3. Empirical selection probability of each variable by stability selection in model 1.** The closer the value on the x-axis is to 1, the more stably the variable is selected across repeated resampling, indicating a high likelihood of association with the target variable. Variables like 'int01x02' represent interaction term of the variable 01 and 02. Please refer to Supplementary Table S3 for the variable numbers and their definitions.

**Table 3. Multivariable Logistic Regression Models with L1 regularization for VPI Prediction.**

| Selected variables | Model 1[a] | Model 2[b] and 3[c] |
|---|---|---|
| | Odds ratio | Odds ratio |
| **logit. Pleural contact** | 1.804 | 1.824[d] |
| **logit. Solid nodule** | 3.001 | 1.522[d] |
| **volume nodule (%)** | | 2.703[d] |
| **Model performance** | ROC-AUC (95% CI) | ROC-AUC (95% CI) |
| **Training cohort** | 0.816 (0.772-0.861) | 0.831 (0.792-0.870) |
| **Test cohort** | 0.782 (0.707-0857) | 0.767 (0.691-0.843) |

[a]Model 1: 22 radiological features by AI analysis

[b]Model 2: 22 radiological features by AI analysis + 3D imaging features

[c]Model 3: 22 radiological features by AI analysis + 3D imaging features + clinicopathological factors

[d]Identical variables were selected in Models 2 and 3.

under the curves (ROC-AUCs) calculated for the test cohort. "Pleural contact" and "Solid nodule" were positively associated with VPI in all models, and the percentage of solid nodule volume also showed a high odds ratio.

In the training cohort, Model 1 was slightly inferior to the more complex Models 2 and 3 (ROC-AUC 0.816 vs. 0.831). However, in the test cohort, Model 1 demonstrated better performance than Models 2 and 3 (ROC-AUC 0.782 vs. 0.767). Based on the results suggesting that the benefits of adding 3D radiological features and clinicopathological factors are limited, Model 1 adopted for the following analysis.

### Performance of AI-derived 22 radiological features in VPI detection

Fig 4 shows the performance of the predictive algorithm for VPI detection, as demonstrated by the ROC curves of Model 1 for both the training and test cohorts. In the training cohort, the model achieved an AUC of 0.816 (95% CI: 0.772–0.861),

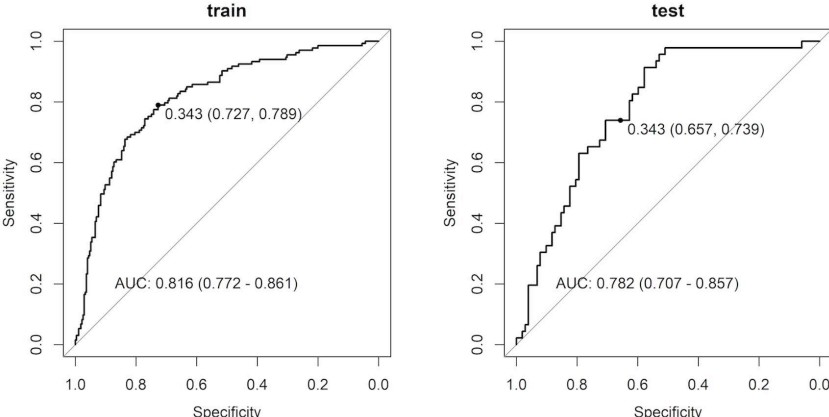

**Fig 4. Predictive performance of model 1 was evaluated based on the ROC curve and its area under the curve (AUC), as well as the sensitivity and specificity at the optimal cutoff.** The cutoff value of 0.343 was the point in the training cohort where the sum of sensitivity and specificity was the highest, and the sensitivity and specificity in the test cohort were also evaluated using the same cutoff.

signifying a robust capability to differentiate between VPI-positive and VPI-negative cases. The VPI prediction probability was calculated using the following formula:

$$\textbf{Prob}.\,(\textbf{VPI}) \;=\; \frac{1}{1 + e^{\textbf{1.0479}\,-\,\textbf{0.590}\times\textbf{logit}(\text{solid})-\textbf{1.099}\times\textbf{logit}(\text{pleural contact})}}$$

The optimal cutoff identified on the curve corresponded to a score of 0.343, yielding a sensitivity of 0.727 and a specificity of 0.789. In the test cohort, the AUC was 0.782 (95% CI: 0.707–0.857) with the same threshold score of 0.343, resulting in a sensitivity of 0.739 and specificity of 0.657. These findings indicate that the model maintained a commendable level of predictive accuracy when applied to new data, even though the test cohort was not necessarily identical to the training cohort. Performance on the leading variable, "Solid nodule," was worse in the test population compared to the training set, with AUC 0.741 (95% CI: 0.658–0.824) versus AUC 0.786 (95% CI: 0.741–0.831), which could have influenced the results in the Supplementary S3 Table. The ROC curves of models 2 and 3 are demonstrated in Supplementary S3 Fig.

Representative cases of the predictive ability of the AI models for VPI are shown in Supplementary S4 Fig. As described above, our VPI prediction formula was computed based on the confidence scores of the two radiological features ("Solid nodules"and"Pleural Contact) identified by the AI. Case (a) had high values for both "Solid nodules" and "Pleural Contact; thus, VPI was positive (Supplementary S4 Figa). Conversely, in case (b), the "Solid nodule" score is high, but the "Pleural Contact" value is low, resulting in a negative VPI (Supplementary S4 Figb). Case (c) is characterized by a low "Solid nodule" score and a high "Pleural Contact" score, which is negative for VPI (Supplementary S4 Figc).

## Discussion

This study pioneered the application of AI to analyze 3D radiological features for predicting VPI in early-stage NSCLC. By utilizing an innovative combination of factors derived from the 3D CT image analysis, our model accurately identified the presence of VPI. Notably, the "Solid nodule" and "Pleural contact" features were pivotal in our analysis, which helped improve the prediction accuracy, with AUCs of 0.816 for the training cohort and 0.782 for the test cohort.

Previous studies have reported the relationship between VPI and poor prognosis in NSCLC [1,2,33]. Notably, pioneering studies, including the influential work by Travis et al., have clearly defined VPI using elastic fiber staining, as described in the current TNM classification, and have emphasized its prognostic significance [34]. VPI is associated with more

aggressive tumors and a higher frequency of lymph node metastases, emphasizing the need for accurate preoperative identification of VPI [3,6]. Although VPI diagnosis is based on pathological findings, predicting VPI through preoperative imaging is challenging. Therefore, advances in preoperative VPI diagnosis are essential for determining surgical strategies and improving clinical outcomes in NSCLC.

Various studies have reported CT image characteristics for predicting VPI. The structure of the tumor attached to the pleura has been identified as a key characteristic. Researchers have emphasized the importance of measuring the tumor's contact length with the pleura using CT scan tools and categorizing its relationship [8,9,35]. Benedikt et al. noted that VPI was more prevalent in solid nodules than in subsolid nodules among surgically resected lung adenocarcinomas [15]. Furthermore, the pleural contact length of this solid area has a more significant correlation with VPI than the pleural contact length of the entire tumor [36]. These findings support our findings, underscoring that solid components and tumor contact are pivotal factors associated with VPI. While previous research has emphasized pleural indentation and tags as significant predictors of VPI, our study did not reveal these factors as substantial predictors [7]. The difference in our findings could be attributed to our study population, which consisted of patients with tumors attached to the pleural surface. It is possible that pleural tags will show a notable association with VPI if we focus only on tumors without pleural contact.

This study highlights the advantages of AI in providing objective, reproducible, and accurate measurements from CT images, independent of variations in physicians' interpretations. Our approach utilizes the advantages of 3D image analysis and overcomes the limitations of 2D-CT slice assessment. Unlike 2D analysis, which assesses only selected slices, 3D analysis evaluates the entire tumor volume in three dimensions, enabling accurate characterization of its overall morphology and spatial relationship to the pleura. Moreover, the AI software beta version of Synapse Vincent system calculates total and solid component volumes and extracts 22 imaging features from 3D reconstruction images using three modules: bronchopulmonary segment prediction, nodule classification, and finding description generation. These metrics cannot be reliably obtained from single-slice 2D-CT [26]. For VPI prediction, it is essential to differentiate between a single area of pleural attachment and an extended area of attachment, because the latter has a greater risk of pleural invasion [35]. In addition, our study analyzed three models that included a multitude of variables ranging from clinical data and 3D tumor volume measurements to 22 radiological features. Multivariate analysis using stepwise, backward, and forward selection yielded different results, so we applied stability selection, which identifies robust predictors in a multivariate logistic regression framework. Two features, "Solid nodule" and "Pleural contract," remained. Our findings led us to explore the crucial factors of AI features in VPI prediction and select Model 1, which provides an optimal balance between simplicity and predictive power. The use of advanced statistical methods to validate and test the model ensured the reliability and applicability of the results. Moreover, the beta version of our AI software [21–23], which is under development for non-commercial use, has proven to be simple and reliable, promising advancements in the field of radiological diagnostics.

This study has several limitations. First, because this was a retrospective study conducted at a single institution, patient selection bias was inevitable. Second, a decrease in performance from the training cohort to the test cohort was observed, despite the high accuracy of our predictive model. Third, there was no direct correlation between the location of VPI on the radiological image and its pathological correspondence, leading to potential discrepancies. Fourth, the "black box" aspect of AI requires a more transparent understanding of its predictive capabilities. However, we excluded lesions not in close contact with the pleura to avoid an artificially high VPI detection accuracy, reflecting the true complexity of identifying VPI in ambiguous clinical scenarios and reinforcing the applicability of our findings in real clinical situations. In tumors adjacent to the pleura, the primary predictors were identified as "pleural contact" and "solid nodule." While pleural indentation and spiculation are typically regarded as indicators of VPI, they did not consistently demonstrate its presence in this setting. Our approach of using AI for 3D image analysis to predict VPI is a robust platform for future developments in NSCLC diagnostics.

In conclusion, AI-assisted analysis of 3D CT imaging enables the preoperative prediction of VPI in NSCLC. "Solid nodule" and "Pleural contact" in CT imaging were two critical factors associated with VPI. Our VPI prediction model predicts

VPI with high accuracy. We believe that our study offers significant insights into VPI prediction using AI-assisted analysis of 3D CT imaging, warranting further research in this area.

## Supporting information

**S1 File.  Supplementary Methods.**
(DOCX)

**S1 Fig.  Examples of logit transformation.** This is the Fig S1 legend. An example of the distribution of AI confidence scores before and after the logit transformation is shown. Before the transformation, the distribution is skewed towards 0 or 1, and it is assumed that there is a nonlinear relationship with the target variable due to ceiling or floor effects. After the logit transformation, the variable follows a more spread-out distribution.
(TIFF)

**S2a Fig.   Selection probabilities by the stability selection of model 2.** This is the fig S2a legend. Empirical selection probability of each variable by stability selection in model 2. The closer the value on the x-axis is to 1, the more stably the variable is selected across repeated resampling, indicating a high likelihood of association with the target variable. Variables like 'int01x02' represent the interaction term of the variables 01 and 02. Please refer to Appendix Table 3 for the variable numbers and their definitions.
(TIFF)

**S2b Fig.   Selection probabilities by the stability selection of model 3.** This is the fig S2b legend. Empirical selection probability of each variable by stability selection in model 3. The closer the value on the x-axis is to 1, the more stably the variable is selected across repeated resampling, indicating a high likelihood of association with the target variable. Variables like 'int01x02' represent the interaction term of the variables 01 and 02. Please refer to Appendix Table 3 for the variable numbers and their definitions.
(TIFF)

**S3 Fig.  Receiver operating characteristics curves of models 2 and 3 in the training and test cohort.** This is the fig S3 legend. The predictive performance of models 2 and 3 was evaluated based on the ROC curve and its area under the curve (AUC), as well as the sensitivity and specificity at the optimal cutoff. The cutoff value of 0.399 was the point in the training cohort where the sum of sensitivity and specificity was the highest, and the sensitivity and specificity in the test cohort were also evaluated using the same cutoff.
(TIFF)

**S4 Fig.  This is the CT images of representative cases with high AI confidence scores of "Pleural contact" and "Solid nodule".** This is the fig S4 legend. Case (a): AI confidence score of "Pleural contact" was 0.99, and "Solid nodule" was 1. This case represented a high "Solid nodule" and "Pleural contact," showing a high VPI probability score using the prediction model. As predicted, the VPI was positive. Case (b): AI confidence score of "Pleural contact" was 0.34, and "Solid nodule" was 0.908. This case represented a VPI negative case with a high "Solid nodule" score and a low "Pleural Contact" score. The prediction model showed a low VPI probability score. Case (c): AI confidence score of "Pleural contact" was 0.99, and "Solid nodule" was 0.14. This case represented a VPI negative situation with a low "Solid nodule" score and a high "Pleural Contact" score. The prediction model showed a low VPI probability score.
(TIFF)

**S5 Fig.  This is the CT images of representative examples of 22 AI-derived radiological features used in the analysis.** This is the fig S5 legend. This figure illustrates representative CT images demonstrating the 22 radiological features

automatically extracted by the AI model. Each panel corresponds to one feature, with visual examples defining each characteristic.
(TIFF)

**S1 Table. This is the Clinicopathological factors and 3D imaging features.**
(DOCX)

**S2 Table. This is the Association between clinicopathologic factors and VPI in the training cohorts.**
(DOCX)

**S3 Table. Relationship between VPI and AI-Derived 22 Radiological Features in the test cohort.**
(DOCX)

## Acknowledgments

We would like to express our gratitude to Mami Murakami for her technical assistance in organizing the data for this study.

## Author contributions

**Conceptualization:** Wakako Nagase, Kazuharu Harada, Yujin Kudo, Masataka Taguri, Norihiko Ikeda.

**Data curation:** Wakako Nagase, Yujin Kudo, Ikki Takada, Jinho Park, Kotaro Murakami, Tatsuo Ohira, Toshitaka Nagao.

**Formal analysis:** Kazuharu Harada, Masataka Taguri.

**Funding acquisition:** Norihiko Ikeda.

**Investigation:** Wakako Nagase, Yujin Kudo.

**Methodology:** Wakako Nagase, Kazuharu Harada, Yujin Kudo, Masataka Taguri, Norihiko Ikeda.

**Project administration:** Yujin Kudo, Masataka Taguri, Norihiko Ikeda.

**Resources:** Yujin Kudo, Jun Matsubayashi, Ikki Takada, Jinho Park, Kotaro Murakami, Tatsuo Ohira, Toshitaka Nagao.

**Supervision:** Yujin Kudo, Jun Matsubayashi, Tatsuo Ohira, Toshitaka Nagao, Masataka Taguri, Norihiko Ikeda.

**Validation:** Kazuharu Harada, Masataka Taguri.

**Visualization:** Wakako Nagase, Kazuharu Harada.

**Writing – original draft:** Wakako Nagase, Kazuharu Harada.

**Writing – review & editing:** Yujin Kudo, Jun Matsubayashi, Ikki Takada, Jinho Park, Kotaro Murakami, Tatsuo Ohira, Toshitaka Nagao, Masataka Taguri, Norihiko Ikeda.

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
