## [Decision Letter · Decision Letter 0]

30 Jun 2025

Dear Dr. Kudo,

Thank you for submitting your manuscript to PLOS ONE. After careful consideration, we feel that it has merit but does not fully meet PLOS ONE’s publication criteria as it currently stands. Therefore, we invite you to submit a revised version of the manuscript that addresses the points raised during the review process.

We look forward to receiving your revised manuscript.

Kind regards,

Jun Hyeok Lim, M.D.

Academic Editor

PLOS ONE

Journal Requirements:

 “Norihiko Ikeda reports receiving a research grant from the FUJIFILM Corporation for the Department of Surgery, Tokyo Medical University.”

“Norihiko Ikeda reports receiving a research grant from the FUJIFILM Corporation for the Department of Surgery, Tokyo Medical University. Although we have utilized the AI software integrated within the Synapse Vincent System (Fujifilm Corporation, Japan) for this study, it is essential to note that this is a development model and not a commercially available product. We have no commercial conflicts of interest to disclose.”

5. In the online submission form you indicate that your data is not available for proprietary reasons and have provided a contact point for accessing this data. Please note that your current contact point is a co-author on this manuscript. According to our Data Policy, the contact point must not be an author on the manuscript and must be an institutional contact, ideally not an individual. Please revise your data statement to a non-author institutional point of contact, such as a data access or ethics committee, and send this to us via return email. Please also include contact information for the third party organization, and please include the full citation of where the data can be found.

6. Please upload a copy of Figure 3 and 4, to which you refer in your text on page 25 and 28. If the figure is no longer to be included as part of the submission please remove all reference to it within the text.

Reviewers' comments:

Reviewer's Responses to Questions

**Comments to the Author**

1. Is the manuscript technically sound, and do the data support the conclusions?

Reviewer #1: Yes

Reviewer #2: No

Reviewer #3: Yes

Reviewer #4: Yes

2. Has the statistical analysis been performed appropriately and rigorously?

Reviewer #1: Yes

Reviewer #2: No

Reviewer #3: Yes

Reviewer #4: No

3. Have the authors made all data underlying the findings in their manuscript fully available?

Reviewer #1: Yes

Reviewer #2: No

Reviewer #3: Yes

Reviewer #4: No

4. Is the manuscript presented in an intelligible fashion and written in standard English?

Reviewer #1: Yes

Reviewer #2: Yes

Reviewer #3: Yes

Reviewer #4: Yes

Reviewer #1: Summary

The authors have developed and validated a model to detect visceral pleural invasion (VPI) in early-stage non-small cell lung cancer (NSCLC) using AI-assisted 3D CT imaging, employing a commercial software, which is easily accessible. The model demonstrates promising diagnostic performance (AUC nearly 0.8) based on features such as "solid nodule" and "pleural contact." However, several issues must be addressed to improve both scientific rigor and clinical applicability.

Major Strengths

The study presents a technically robust approach to a clinically meaningful problem. By focusing on AI-assisted 3D CT imaging, the authors overcome limitations inherent to conventional 2D imaging and subjective radiological interpretation. In particular, 3D deep learning-derived features of various cancers can be the “next radiomics” features as they are now easily implemented and accessible not only across cancer types but also across modality (i.e. CT and MR). It would be appreciated if the authors consider to cite the following paper by Lee and Ahn et al. (Neuro-Onc, 2024) as one of the previous studies of AI-driven 3D imaging features for other types of cancers: https://pubmed.ncbi.nlm.nih.gov/37855826/.

The use of objective, algorithmically extracted features such as "solid nodule" and "pleural contact" enhances reproducibility, and the application of stability selection and regularized logistic regression adds methodological rigor.

Major Weaknesses

Despite the strong methodological framework, several limitations reduce the immediate clinical applicability of the model.

The lack of external validation raises concerns about generalizability across institutions and imaging settings. The authors are encouraged to do external validation or if not possible, at least 5-fold cross validation to provide the generalized performance of the developed model.

The model was developed exclusively on tumors‚ larger than 4 cm and in pleural contact, potentially introducing selection bias and limiting applicability to the broader NSCLC population. Please provide the references to justify this exclusion criteria.

Table 2 seemed to provide the comparison with radiologist performance or assessment with the model performance. However, these results can be more emphasized and elaborated in the Discussion.

Minor Issues

The resolution of Figure 1B, which is crucial, is too low to recognize. Please submit the figure as high resolution as possible. Are there any attention maps such as Grad-CAM, which is provided by the software?

The term "pleural contract" appears to be a typographical error and should be corrected to "pleural contact" (e.g., Table 2, page 20).

Recommendation

This is a valuable and technically well-executed study with clear clinical potential. However, given the aforementioned limitations-particularly the lack of external validation, unclear clinical integration, and limited generalizability-a Major Revision is recommended.

Reviewer #2: The significant differences in age and pathological tumor size observed between the training and test cohorts highlight the need for cross-validation to confirm the robustness and generalizability of the developed model.

The statistical analysis presented in Table 2 appears incomplete. It is recommended to perform both univariate and multivariate logistic regression analyses, complemented by AUC values, to identify and validate independent predictors of vascular invasion (VPI).

The table titles in the manuscript lack clarity. To improve understanding:

Table 1 should explicitly specify that it compares baseline characteristics between the cohorts, e.g., "Baseline Characteristics: Training versus Test Cohorts."

Table 3 should have a more descriptive title, such as "Multivariable Logistic Regression Models for VPI Prediction."

A DeLong test should be conducted to perform a statistical comparison between the ROC curves derived from the training and test cohorts, providing a quantitative assessment of their differences.

In the introduction section (lines 61–63), the critique regarding the limitations of 2D CT imaging would benefit from citing comparable 3D AI studies. Including and cite references such as the study with DOI: 10.1016/j.crad.2021.11.008 would strengthen the literature context.

Volumetric parameters, such as the solid-part volume ratio, show high odds ratios but were excluded from the final predictive model without justification. Clarifying the rationale for their exclusion is essential.

The manuscript mentions 22 radiological features derived via AI; however, these features are not listed. Including a figure and detailed description of these features would enhance clarity and facilitate reproducibility.

Reviewer #3: This is a well-written study addressing the potential of AI-based 3D CT imaging for preoperative prediction of visceral pleural invasion (VPI) in early-stage NSCLC. The performance of the AI model is encouraging, with AUC values above 0.75 in both cohorts.

I have the following concerns.

1. Please revise “spicula” throughout the manuscript to the correct term “spiculation.”

2. Partial resection” should be corrected to the standard term “wedge resection.” This improves clinical clarity and aligns with surgical literature conventions.

3. The clinical impact of preoperative VPI detection is insufficiently justified. Since VPI is ultimately diagnosed on pathology, the value of preoperative prediction needs clearer articulation. Please explain why such prediction meaningfully changes surgical planning or postoperative management, especially considering that adjuvant treatment decisions can be made after pathological evaluation.

4. The manuscript states that 3D AI imaging offers advantages over 2D but does not clearly explain what is gained by the 3D approach. Please clarify which specific features or metrics benefit from 3D analysis, and why they cannot be captured in 2D AI-based models.

5. The distinction between “round shape” and “smooth shape” is unclear. Are these mutually exclusive? Are they referring to surface regularity versus contour circularity? Please define these features precisely, preferably with visual examples or definitions.

6. The term “fatness” in Table 2 and results is unclear. Does this refer to fat-containing nodules or another imaging characteristic? Please clarify or revise this term for precision.

7. Please include an explanation in the text for why “Pleural contact” and “Solid nodule” were chosen over others such as “spiculation” or “pleural indentation,” which are commonly associated with VPI.

Reviewer #4: The authors present an interesting application of AI-driven 3D CT analysis for predicting visceral pleural invasion (VPI) in early-stage NSCLC. The authors should address the overfitting concerns through more rigorous validation and contextualize their work within the broader landscape of AI applications in lung cancer imaging.

The decrease in AUC from training (0.816, 0.831) to test (0.782, 0.767) cohort, combined with the drop in the key 'Solid nodule' feature performance (0.786→0.741), suggests potential overfitting that could limit clinical generalizability. Common solutions are 1) implement cross-validation within the training set to better estimate true model performance and detect overfitting earlier. 2) consider ensemble methods or additional regularization techniques to improve model stability, 3)report confidence intervals for all performance metrics and test for statistical significance of performance differences. Among these approaches, implementing cross-validation (option 1) is particularly essential and would substantially strengthen the validation framework.

The manuscript would also benefit from broader contextualization within the growing field of deep learning applications for lung cancer CT imaging. Given that both studies employ deep learning on CT imaging for lung cancer applications, citing 'Improving lung cancer diagnosis and survival prediction with deep learning and CT imaging' would provide valuable context for how AI-driven approaches are advancing across different aspects of lung cancer management, from diagnosis to prognosis to surgical planning.

**Do you want your identity to be public for this peer review?** For information about this choice, including consent withdrawal, please see our Privacy Policy

Reviewer #1: No

Reviewer #2: No

Reviewer #3: No

Reviewer #4: No

---

## [Author Response · Author response to Decision Letter 1]

18 Aug 2025

Please refer to the separately attached document for our detailed, point-by-point responses to all specific reviewer and editor comments.

---

## [Decision Letter · Decision Letter 1]

8 Sep 2025

AI-driven 3D CT imaging prediction model for improving preoperative detection of visceral pleural invasion in early-stage lung cancer

PONE-D-25-21329R1

Dear Dr. Kudo,

We’re pleased to inform you that your manuscript has been judged scientifically suitable for publication and will be formally accepted for publication once it meets all outstanding technical requirements.

Kind regards,

Jun Hyeok Lim, M.D.

Academic Editor

PLOS ONE

Additional Editor Comments (optional):

Reviewer #1:

Reviewer #2:

Reviewer #4:

Reviewers' comments:

Reviewer's Responses to Questions

**Comments to the Author**

Reviewer #1: All comments have been addressed

Reviewer #2: (No Response)

Reviewer #4: All comments have been addressed

2. Is the manuscript technically sound, and do the data support the conclusions?

Reviewer #1: Yes

Reviewer #2: (No Response)

Reviewer #4: Yes

3. Has the statistical analysis been performed appropriately and rigorously?

Reviewer #1: Yes

Reviewer #2: (No Response)

Reviewer #4: Yes

4. Have the authors made all data underlying the findings in their manuscript fully available?

Reviewer #1: No

Reviewer #2: (No Response)

Reviewer #4: No

5. Is the manuscript presented in an intelligible fashion and written in standard English?

Reviewer #1: Yes

Reviewer #2: (No Response)

Reviewer #4: Yes

Reviewer #1: The authors have adequately addressed my comments. Clarifications on validation strategy and selection criteria were well incorporated with appropriate references. The discussion of model performance and statistical methods was strengthened, and the typographical and figure quality issues were corrected. Overall, the revisions appropriately reflect the concerns raised and improve the clarity and rigor of the manuscript.

Reviewer #2: After revision, the manuscript has shown considerable improvement. Given these enhancements, I recommend the manuscript for acceptance.

Reviewer #4: Thank you for your comprehensive response. Your explanations regarding the temporal validation approach, regularization through stability selection, and focus on model interpretability are well-reasoned. I accept your methodological choices and appreciate the clinical considerations that guided your study design.

**Do you want your identity to be public for this peer review?** For information about this choice, including consent withdrawal, please see our Privacy Policy

Reviewer #1: No

Reviewer #2: No

Reviewer #4: No

---

## [Editor Report · Acceptance letter]

PONE-D-25-21329R1

PLOS ONE

Dear Dr. Kudo,

I'm pleased to inform you that your manuscript has been deemed suitable for publication in PLOS ONE. Congratulations! Your manuscript is now being handed over to our production team.

Kind regards,

on behalf of

Dr. Jun Hyeok Lim

Academic Editor

PLOS ONE